# Potential Benefits of a Single Session of Remote Ischemic Preconditioning and Walking in Sedentary Older Adults: A Pilot Study

**DOI:** 10.3390/ijerph20043515

**Published:** 2023-02-16

**Authors:** Elena Muñoz-Gómez, Sara Mollà-Casanova, Núria Sempere-Rubio, Pilar Serra-Añó, Marta Aguilar-Rodríguez, Diego A. Alonso-Aubin, Iván Chulvi-Medrano, Marta Inglés

**Affiliations:** 1Research Unit in Clinical Biomechanics (UBIC), Department of Physiotherapy, Faculty of Physiotherapy, University of Valencia, 46010 Valencia, Spain; 2Strength Training and Neuromuscular Performance Research Group (STreNgthP), Faculty of Health, Camilo José Cela University, 28692 Madrid, Spain; 3UIRFIDE Research Group, Physical Education and Sport Department, University of Valencia, 46010 Valencia, Spain

**Keywords:** ischemic conditioning, cardiovascular health, physical function, elderly, endurance exercise

## Abstract

Ischemic preconditioning (IPC) has shown positive effects in endurance-type sports among healthy young individuals; however, its effects in endurance-type exercises in older adults have not been explored. We aimed to examine the acute effects of a single session of IPC prior to an endurance-type exercise on cardiovascular- and physical-function-related parameters in sedentary older adults. A pilot study with a time-series design was carried out. Nine participants were enrolled consecutively in the following intervention groups: (i) SHAM (sham IPC + walking) and (ii) IPC (IPC + walking) groups. The main outcomes were resting systolic (SBP) and diastolic (DBP) blood pressure, heart rate (HR), peripheral oxygen saturation (SpO_2_), maximum isometric voluntary contraction (MIVC), endurance performance, and perceived fatigue. After the intervention, the IPC group showed a significant reduction in SBP, whereas SpO_2_ decreased in the SHAM group. The IPC group maintained quadriceps MIVC levels, whereas these levels dropped in the SHAM group. No changes in DBP, resting HR, endurance, or fatigue in any group were observed. These findings are of interest for the promotion of cardiovascular and physical health in older people.

## 1. Introduction

Ischemic preconditioning (IPC) is a non-invasive procedure that involves transient cycles of arterial blood flow occlusion and reperfusion. This technique was initially developed to induce repetitive, sublethal ischemia bursts prior to an ischemic threat, thus improving tissue tolerance and offering subsequent protection against such events [1]. Although IPC intervention was first demonstrated to protect the heart against acute myocardial infarction when applied locally, its beneficial effects have also been reported when applied to other organs and tissues, such as the kidney, intestine, lung, liver, brain, and skeletal muscle (reviewed in Hausenloy and Lim [2]), thus providing evidence for its systemic effects.

These effects prompted the sports scientific community to speculate that IPC would also exert beneficial effects on exercise performance, and IPC was thus translated to the sports setting, by using a standard blood pressure cuff placed on the upper arm or leg [3]. Indeed, IPC applied in the absence of concurrent exercise or therapy has shown positive effects in different endurance-type sports modalities, such as cycling, running, and swimming, with greater effects observed in healthy, untrained individuals [4,5].

The precise mechanisms through which IPC might induce such ergogenic effects remain unclear; however, several findings suggest that its positive effect on endurance performance may be due to increased local vasodilation [6], blood flow [7], and O_2_ uptake [8], as well as decreased lactate accumulation [9].

However, these studies have been carried out in healthy young individuals, mostly athletes, whereas its precise effect on the sedentary elderly population remains unclear. This is of interest since it has been posited that the physiological responses to IPC are age- and sex-dependent [10] and are related to the individual’s training level [4].

Aside from the sports setting, a single session of IPC has been shown to reduce both the systolic (SBP) and the diastolic (DBP) blood pressure in normotensive middle-aged adults [11]. Regarding functional parameters, a single session of IPC has been reported to improve strength [12] in stroke survivors, endurance capacity in people with multiple sclerosis [13], and handgrip strength and functional capacity in active elderly women [14].

However, to the best of our knowledge, all these studies have investigated the effects of a single bout of IPC on cardiovascular and functional aspects in the absence of consecutive exercise, but there are no studies investigating such effects after a traditional resistance exercise, such as walking.

In view of the above, the aim of the current study was to investigate the effect of a single intervention session combining IPC and walking on cardiovascular parameters and physical function in sedentary older adults.

## 2. Materials and Methods

### 2.1. Participants

Nine older adults were enrolled in the study (mean age 69.11 (3.10) years). They were recruited through advertisements and social media from July to September 2021 to participate in the study. Inclusion criteria were being aged between 65 and 90 years; being physically inactive (<150 min of physical exercise per week); and having signed an informed consent form. Exclusion criteria were the following: (i) institutionalized patients; (ii) history of stroke in the last 6 months or hospital admission for any reason in the last 3 months; (iii) uncontrolled hypertension; (iv) medicated with anticoagulants; (v) oncological patient under active treatment: chemotherapy or radiotherapy; (vi) neurological or musculoskeletal impairments limiting the capacity to undertake the exercise and testing requirements of the study; (vii) cognitive impairment (score below 25 on the “Mini-Mental Test”, conducted by a qualified physiotherapist); or (viii) severe disability (score below 15 points on the Barthel scale).

All the assessments were carried out in a research laboratory of the Department of Physiotherapy at the University of Valencia.

All individuals were informed verbally and in writing in plain language about the methods, procedures, and potential risks of participation. Written informed consent was obtained from all the participants included in the study. All study procedures were approved by the institutional Ethical committee at the University of Valencia (registration number 1855485) and were carried out in accordance with the principles of the World Medical Association’s Declaration of Helsinki.

### 2.2. Experimental Design

This time-series study was designed to examine the effect of a single intervention session combining IPC and walking on cardiovascular parameters and physical function in sedentary older adults. The participants underwent two consecutive interventions (a 2-week “washout” period in between): (i) SHAM (sham ischemic preconditioning + walking at a moderate intensity) and (ii) IPC (3 ×5 min ischemic preconditioning at 50 mmHg above systolic blood + walking at a moderate intensity). This study design allowed each participant to serve as his or her own control avoiding the common heterogeneity of the target population. This was a single-blind study, as the participants did not know whether they were receiving the sham or the IPC intervention. The participants were assessed at baseline (T0) and after (T1) intervention for resting systolic (SBP) and diastolic (DBP) blood pressure, heart rate (HR), peripheral oxygen saturation (SpO_2_), lower-extremity maximum isometric voluntary contraction (MIVC), endurance performance, and perceived fatigue.

### 2.3. Procedure

As stated before, the participants were consecutively allocated to the two intervention groups following a counterbalanced order: (i) SHAM (sham ischemic preconditioning + walking) and (ii) IPC (ischemic preconditioning + walking), as explained below. During each session, potential harms or adverse effects were recorded. The washout period between sessions lasted two weeks. The individuals were asked to be committed to avoiding any training program during the washout period (they would be excluded if reporting practicing any physical activity). To analyze the effect of the interventions, two assessments were performed: one before treatment (T0) and the other one at end of each intervention (T1). Figure 1 schematically represents the procedure.

(IPC) IPC + walking: Ischemic preconditioning (IPC) was performed using bilateral arterial occlusion with the participant lying in a supine position. Occlusion cuffs (96 × 13 cm; Komprimeter Riester, Jungingen, Germany) were placed bilaterally on the upper thigh of the participant. For the occlusion phase, the cuff was inflated to 50 mmHg above the systolic blood pressure for 5 min, which is sufficient to obstruct blood flow, as previously shown [15]. This ischemic procedure was repeated three times, alternated with 5 min of reperfusion (i.e., 3 × 5 min occlusion/5 min reperfusion). Immediately after, the participants were asked to walk 20 min at a moderate intensity (i.e., 65–75% HR_max_, 12–13 Borg RPE scale) [16] on an open field circuit. The individuals’ heart rate was monitored using a Polar HR monitor with a Pro Strap (Polar Electro Oy, Kempele, Finland) to ensure individualized target heart rate parameters, which were calculated according to the Karvonen formula [17]. The individuals were then asked to continue walking for 5 min but at a slower pace and lower intensity to cool down.

(SHAM) Sham IPC + walking: the individuals underwent the same intervention as that described above, but the cuff was inflated only to 20 mmHg to serve as a placebo, as previously recommended [15].

Both for the SHAM and for the IPC interventions, the cuffs were applied simultaneously, but the pressure was applied alternatively; that is, when the cuff was inflated on one thigh, the other was under reperfusion, thus making a total intervention time of 30 min. Furthermore, to normalize the cognitive appraisal of both interventions and avoid any potential nocebo effect [18], the participants were informed that both external pressure conditions were tested, as they could improve performance.

### 2.4. Outcomes

To assess the impact of the intervention on cardiovascular parameters, the resting systolic blood pressure (SBP), the diastolic blood pressure (DBP), and the heart rate (HR) were determined using an Omron HEM-7201 upper arm automatic blood pressure monitor (Omron Healthcare, Kyoto, Japan). The participants had to remain seated in a chair before measurement for 10 min. Peripheral oxygen saturation levels were non-invasively measured using a pulse oximeter (Quirumed, Valencia, Spain), placed on the index finger. This device detects the amount of oxyhemoglobin and deoxygenated hemoglobin in arterial blood and shows oxyhemoglobin saturation (SpO_2_), which is an indirect estimation of arterial oxygen saturation (SaO_2_) [19].

Regarding the physical function parameters, we evaluated the maximum isometric voluntary contraction (MIVC) of the main lower limb muscles involved in gait, such as the gluteus medius, quadriceps, hamstrings, tibialis anterior, and triceps surae muscles by means of a force sensor connected to the relevant software (Chronojump Boscosystem, version 2.1.2). Before performing data acquisition, the force sensor was calibrated in the position in which each exercise was to be performed. The participant was asked to exert maximum force as fast as possible and to hold the maximum force for 5 s. This procedure was repeated 3 times, with 5 s rests between them. We calculated the mean and maximal force of the three measurements, using the three middle seconds (i.e., seconds 2, 3, and 4) of each repetition. The order of muscle groups was randomized among the patients. When repetitions differed by more than 20%, these were discarded, and another measurement was registered.

Furthermore, endurance performance was assessed by determining the distance covered in the six-minute walk test (6MWT). The participants were asked to walk as fast as possible, without running, along a 30 m hallway for 6 min. They were allowed to take as many standing rests as necessary, but the timer kept going. This test has shown an excellent test–retest reliability (ICC = 0.95) in community-dwelling elderly people [20].

Finally, self-reported fatigue was recorded after the intervention on the 20-point Borg Rating of Perceived Exertion (RPE) 6–20-point scale, in which 6 means perceiving “no exertion at all”, and 20 means perceiving a “maximal exertion” of effort. This scale has shown a high test–retest reliability (ICC = 0.93) in older adults [21].

All measurements were conducted by a physiotherapist widely experienced in the measurements of cardiovascular and functional parameters who provided instructions to the participants.

### 2.5. Statistical Analyses

All statistical analyses were performed with SPSS v.24 (IBM SPSS, Inc., Chicago, IL, USA). Standard statistical methods were used to obtain the mean and standard deviation (SD). Inferential analyses of the data were performed using two-way repeated-measure multivariate analysis of variance (MANOVA) with two intra-subject factors: “group” having two categories (SHAM and PAI), and “time” having two categories (T0 and T1). Post hoc analyses were conducted using the Bonferroni correction. The normality and sphericity assumptions were checked using Shapiro–Wilk and Mauchly’s tests, respectively. Type I error was established as ≤ 5% (*p* ≤ 0.05). The effect size was computed using Cohen’s d, thus rating the effect size as follows: large (>0.80), medium (0.50–0.80), or small (0.20–0.50) [22].

## 3. Results

### 3.1. Cardiovascular-Related Parameters

As shown in Table 1, a significant reduction in SBP was observed in the IPC group (*p* = 0.047, d = 0.56), while no significant changes were found in the SHAM group (*p* > 0.05) after the intervention. Furthermore, there was a significant decrease in SpO_2_ in the SHAM group (*p* = 0.05, d = 1.11), whereas SpO_2_ values were maintained in the IPC group (*p* > 0.05) after the exercise. Regarding DBP and resting HR, no changes were observed in either group after the intervention (*p* > 0.05). No intervention-related adverse effects were reported.

### 3.2. Physical-Function-Related Parameters

Upon the analysis of the effects of the interventions on lower-limb isometric strength, a significant decrease in maximal and mean quadriceps MIVC was observed in the SHAM group after the intervention (*p* = 0.04, d = 0.53, and *p* = 0.04, d = 0.53, respectively). However, in the IPC group, both maximal and mean quadriceps MIVC values were similar to baseline values after the intervention (*p* > 0.05 in both cases). The initial values for the gluteus medius, hamstrings, tibialis anterior, and triceps surae muscles were maintained after SHAM and IPC interventions (*p* > 0.05 for all muscle groups) (Table 2).

With regard to endurance performance and fatigue, as measured in terms of the distance covered and perceived fatigue during the 6MWT, respectively, neither the IPC nor the SHAM group improved after the intervention (*p* > 0.05) (Table 2).

## 4. Discussion

This study appraised the effects of a single bout of IPC prior to an endurance-type exercise (i.e., walking) on cardiovascular and physical function parameters in sedentary older adults. We report that a single bout of IPC prior to a moderate-intensity walking session has an impact on cardiovascular parameters (i.e., reduces SBP and helps to maintain SpO_2_ levels but does not affect DBP or HR). IPC further helped to maintain quadriceps MIVC after the intervention but had no effect on endurance performance or perceived fatigue. To the best of our knowledge, this is the first study to investigate the effects of a single IPC session prior to an endurance-type exercise on the aforementioned parameters in sedentary older adults.

The relationship between IPC and blood pressure raised much interest following the work by Madias showing that the acute bouts of IPC in normotensive middle-aged individuals resulted in reduced SBP (>6 mmHg) and DBP (>3 mmHg) [11]. These findings led to suggest that using IPC as a potential therapeutic modality for controlling blood pressure might be appropriate [23]. In agreement with this hypothesis, our results showed that a single session of IPC prior to an endurance-type exercise reduced SBP by 7.12 mmHg but did not affect DBP. This is important since SBP has been known to have a greater effect on BP control than DBP, and therefore its reduction has been used as a therapeutic target [24]. Indeed, although both SBP and diastolic DBP increase with age, after the age of 60 years, SBP continues to rise while DBP declines, probably due to central arterial stiffness, hemodynamic mechanical changes, or even endothelial dysfunction [25]. In addition, we found that the experimental intervention helped to maintain SpO_2_ levels, whereas the sham intervention resulted in a significant decrease in this parameter. This may indicate that the proposed intervention helped to attenuate exercise-induced arterial hypoxemia, probably by enhancing lung volume and, in turn, pulmonary diffusion capacity [26]. Finally, although not statistically significant, a tendency for an HR reduction was found in the experimental group, whereas a tendency for an increase was found in the sham group. A recent systematic review and meta-analysis concluded that there was no significant acute change in HR after an isolated IPC exposure [23]. Since our experimental design included a combined protocol (i.e., IPC + walking), the results are not entirely comparable.

In terms of physical status, it has been shown that short-term IPC interventions increase muscle strength in trained athletes [27]. In addition, handgrip strength significantly improved immediately after an IPC intervention in active elderly women [14]. Furthermore, a single session of IPC can increase strength by 16% in the paretic leg through improved muscle activation in chronic stroke survivors [12]. In line with these results, we found that both mean and maximum quadriceps MIVC significantly decreased in the sham group, but not in the ICP group, for whom these levels were maintained. A possible explanation may be that IPC may enhance the gain of descending excitatory inputs by increasing the excitability of motor-neuron pools, thus improving torque output [12]. However, all the other explored muscles did not experience a decrease in MIVC in the SHAM group, so we cannot extrapolate the effect of IPC to all lower-limb muscles.

Regarding endurance performance and perceived fatigue, our results showed that a single session of IPC before the walking session failed to increase endurance performance as expected, in terms of the distance covered during the 6MWT. Similarly, there was no significant change in the participants from the SHAM group (neither a significant increase nor a decrease in endurance performance, nor perceived fatigue). Therefore, we cannot conclude that IPC exhibited any effects different from those of the SHAM intervention. Similarly, in Telles et al.’s study [14], although the IPC group significantly improved the number of meters in the 6MWT compared with a control group, non-significant differences compared with a sham intervention were found. A recent study in people with multiple sclerosis found that a single cycle of IPC resulted in an immediate improvement of 5.7% in the distance walked during the 6MWT [13]. Nonetheless, they did not perform any endurance-type exercise after the IPC intervention, and therefore no tiredness was expected. Furthermore, we found that the Borg RPE scale before and after intervention or sham was not significantly different, which is consistent with previous studies investigating the acute effects of IPC on fatigue [13]. Further studies should ascertain whether longer treatment periods achieve a positive effect on this variable in older people, as is the case, for instance, with stroke survivors [28].

Overall, our results suggest that a single bout of IPC prior to a moderate-intensity walking session has an impact on cardiovascular parameters (i.e., reduces SBP and helps to maintain SpO_2_ levels but does not affect DBP or HR) and quadriceps MIVC but not on endurance performance or perceived fatigue in sedentary older people.

Our study has some limitations that need to be considered when interpreting the results and/or planning future studies. First, this was a pilot study with a small sample size, so future research is needed to confirm these findings in a larger population. In addition, we did not include a no-treatment control group since our goal was to explore the effect of IPC on a subsequent walking session. Nevertheless, the inclusion of a placebo group as a comparator for IPC has been recently considered the ideal study design for investigating IPC effectiveness [15]. Furthermore, future studies should include longer treatment periods to ascertain whether IPC, when applied chronically, can exert beneficial effects on the studied variables.

## 5. Conclusions

Without waiving any of the limitations listed above, a single session of IPC prior to an endurance-type exercise (i.e., walking) improves cardiovascular parameters (i.e., reduces SBP and helps to maintain SpO_2_ levels) beyond those achieved with a traditional walking exercise. Furthermore, the proposed intervention has a positive effect on quadriceps maximal isometric strength but does not affect endurance performance or perceived fatigue in sedentary older people. Future studies are needed to confirm our findings in a larger population.

## Figures and Tables

**Figure 1 ijerph-20-03515-f001:**
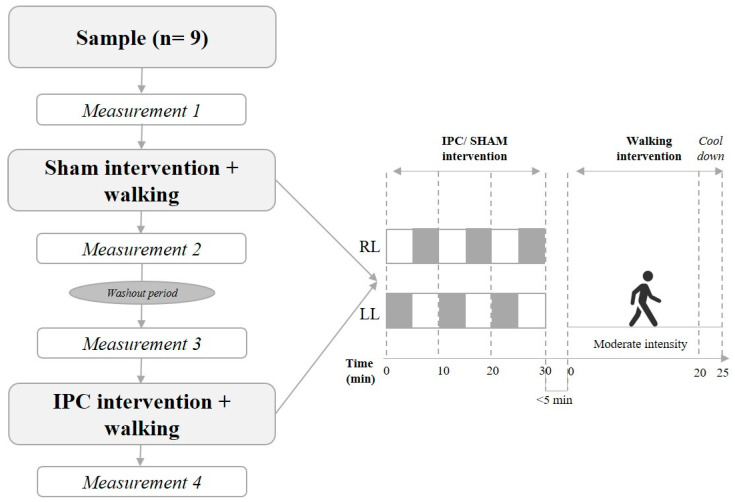
Protocol of the study. IPC = ischemic preconditioning; RL = right leg; LL = left leg. Grey square boxes represent 5 min arterial or sham occlusion, alternated by 5 min reperfusion (white boxes).

**Table 1 ijerph-20-03515-t001:** Cardiovascular parameters before and after the interventions.

	IPC			SHAM		
	T0	T1	T0 − T1 Mean Difference (95% CI)	T0	T1	T0 − T1 Mean Difference (95% CI)
SBP (mmHg)	129.56 (14.62)	122.44 (10.93) *	7.11 (0.12 to 14.1)	131.44 (10.54)	131 (8.83)	0.44 (−7.23 to 8.12)
DBP (mmHg)	71.33 (9.29)	74.33 (8.41)	−3.00 (−7.09 to 1.09)	77.44 (10.61)	77.33 (8.40)	0.11 (−6.68 to 6.90)
HR (bpm)	90.33 (31.99)	83.33 (9.41)	7.00 (−21.66 to 35.66)	78.44 (11.96)	82.44 (7.91)	−4.00 (−14.24 to 6.24)
SpO_2_ (%)	96.56 (2.79)	95.89 (2.42)	0.67 (−0.66 to 2.00)	97.44 (2.13)	94.44 (3.28) *	3.00 (0.00 to 6.00)

Data are shown as mean (standard deviation) and mean differences (95% confidence interval) between T0 and T1. T0: preintervention assessment; T1: postintervention assessment; IPC: ischemic preconditioning; SBP: systolic blood pressure; DBP: diastolic blood pressure; HR: heart rate; SpO_2_: oxyhemoglobin saturation; bpm: beats per minute. *: *p* < 0.05 vs. T0.

**Table 2 ijerph-20-03515-t002:** Effect of the intervention on lower-limb MIVC, endurance performance, and perceived fatigue.

	IPC			SHAM		
	T0	T1	T0 − T1 Mean Difference (95% CI)	T0	T1	T0 − T1 Mean Difference (95% CI)
Gluteus medius maximum MIVC (N)	106.26 (44.22)	102.70 (42.68)	3.55 (−8.06 to 15.17)	104.41 (46.55)	107.22 (52.28)	−2.82 (−29.59 to 23.96)
Gluteus medius mean MIVC (N)	99.00 (43.96)	92.58 (41.58)	6.42 (−6.31 to 19.16)	96.44 (46.76)	101.84 (58.85)	−5.4 (−36.54 to 25.74)
Quadriceps maximum MIVC (N)	219.21 (92.45)	203.33 (77.71)	15.89 (−7.13 to 38.90)	239.20 (106.49)	193.46 (64.74) *	45.73 (2.58 to 88.89)
Quadriceps mean MIVC (N)	204.27 (88.03)	189.71 (75.48)	14.56 (−7.36 to 36.48)	226.04 (103.77)	181.40 (64.40) *	44.64 (3.98 to 85.30)
Hamstrings maximum MIVC (N)	66.18 (29.19)	78.39 (34.37)	−12.21 (−25.06 to 0.64)	64.92 (36.01)	78.06 (62.84)	−13.14 (−52.17 to 25.88)
Hamstrings mean MIVC (N)	59.86 (26.71)	78.66 (49.99)	−18.81 (−45.40 to 7.79)	57.72 (33.35)	71.63 (60.00)	−13.91 (−51.75 to 23.93)
Tibialis anterior maximum MIVC (N)	165.57 (53.73)	159.69 (33.68)	5.88 (−21.73 to 33.49)	155.12 (43.35)	159.88 (51.53)	−4.76 (−39.56 to 30.05)
Tibialis anterior mean MIVC (N)	155.89 (52.57)	149.20 (31.66)	6.69 (−21.04 to 34.42)	141.55 (35.89)	148.45 (50.83)	−6.90 (−39.02 to 25.22)
Triceps surae maximum MIVC (N)	206.00 (52.41)	227.29 (59.65)	−21.29 (−54.42 to 11.83)	185.13 (66.82)	204.23 (96.04)	−19.10 (−71.98 to 33.78)
Triceps surae mean MIVC (N)	179.79 (54.19)	210.96 (53.27)	−31.17 (−64.04 to 1.71)	173.65 (62.67)	191.91 (88.78)	−18.26 (−67.15 to 30.63)
Endurance performance (6mWT, meters)	500.30 (57.85)	496.97 (61.74)	3.33 (−22.56 to 29.23)	500.78 (53.38)	505.41 (66.41)	−4.63 (−26.07 to 16.80)
Perceived fatigue (6–20 Borg RPE scale, points)	11.44 (2.19)	11.00 (2.24)	0.44 (−1.44 to 2.33)	11.00 (2.06)	12.11 (2.09)	−1.11 (−2.76 to 0.54)

Data are shown as mean (standard deviation) and mean differences (95% confidence interval) between T0 and T1. IPC: ischemic preconditioning; T0: preintervention assessment; T1: postintervention assessment; MIVC: maximum isometric voluntary contraction; 6mWT: six-minute walk test; *: *p* < 0.05 vs.T0.

## Data Availability

The data presented in this study are available on request from the corresponding author. The data are not publicly available due to their containing information that could compromise the privacy of research participants.

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
