# Peer review of "Potential Benefits of a Single Session of Remote Ischemic Preconditioning and Walking in Sedentary Older Adults: A Pilot Study"

_ijerph, 2023, doi:10.3390/ijerph20043515_

Round 1

Reviewer 1 Report

Thank you so much for giving me the opportunity to review the article titled “Potential benefits of a single session of remote ischemic preconditioning and walking in sedentary older adults. A pilot study”. The manuscript is written well however few methodological concerns are suggested

No information on allocation randomization for both intervention groups, blinding (single or double); concealment of the participants; data collection setting (laboratory/hospital??)

Who conducted the ‘Mini-mental states’, and what is the qualification of the researchers?  Was there any specialist (medical doctor or physiochemical or physiotherapist or excise medicine) present and validate the intervention parameters (blood pressure or heart rates?), Who provided instructions to the participants?

Who collected data, monitored, and measured the study parameters such as blood pressure (SBP & DBP); heart rate, peripheral oxygen saturation (any biosample?), lower-extremity Maximum Isometric Voluntary Contraction, endurance performance, and perceived fatigue (what was the measuring unit).

How was it confirmed or ensured that the individuals were not performing any training program during the washout period?

Does this study protocol apply for Universal Trial Number under World Health Organization?

Author Response

Response to reviewers: Potential benefits of a single session of remote ischemic preconditioning and walking in sedentary older adults. A pilot study

Manuscript ID: ijerph-2228162

We thank the reviewers (1 and 2) for their critical review and valuable comments. We have taken into account all of their recommendations and suggestions. Itemized responses are listed below. The modifications have been tracked throughout the manuscript to make its revision easier (See Revised Manuscript).

Reviewer: 1

Comments to the Author

Thank you so much for giving me the opportunity to review the article titled “Potential benefits of a single session of remote ischemic preconditioning and walking in sedentary older adults. A pilot study”. The manuscript is written well however few methodological concerns are suggested

  1. No information on allocation randomization for both intervention groups, blinding (single or double); concealment of the participants; data collection setting (laboratory/hospital??)

We thank the reviewer for his/her comments. We have included information on blinding of participants (line 95-96) and data collection setting (line 79-80). However, since it was a series design study, all the participants underwent two consecutive interventions (first, SHAM intervention, and second, IPC intervention), and there was no randomization. Lines 88-95.

  1. Who conducted the ‘Mini-mental states’, and what is the qualification of the researchers? Was there any specialist (medical doctor or physiochemical or physiotherapist or excise medicine) present and validate the intervention parameters (blood pressure or heart rates?), Who provided instructions to the participants?

According to the author’s suggestion, we have specified that the Mini-mental Test was conducted by a qualified physiotherapist. Lines 77-78.

In addition, all measurements and collecting data were conducted by a physiotherapist widely experienced in the measurements of cardiovascular and functional parameters who also provided instructions to the participants. Lines 167-169

  1. Who collected data, monitored, and measured the study parameters such as blood pressure (SBP & DBP); heart rate, peripheral oxygen saturation (any biosample?), lower-extremity Maximum Isometric Voluntary Contraction, endurance performance, and perceived fatigue (what was the measuring unit).

As explained above, the researcher in charge of collecting the data, monitoring and measuring all the parameters was an expert physiotherapist in the field (lines 167-169).

Regarding peripheral oxygen saturation levels were noninvasively measured by a pulse oximeter placed in the index finger (lines 142-146).

With regard to self-reported fatigue, the 20-point Borg Rating of Perceived Exertion (RPE) has no units of measurement, since it is a numerical scale in which 6 means perceiving “no exertion at all” and 20 perceiving a “maximal exertion” of effort. This information has been added on lines 166-167.

  1. How was it confirmed or ensured that the individuals were not performing any training program during the washout period?

As indicated in the lines 108-109, participants were committed not to perform any physical activity during the washout period. Otherwise, they had to inform the researcher and would be excluded from the study. 

  1. Does this study protocol apply for Universal Trial Number under World Health Organization?

We thank the reviewer for his/her appreciation. As we stated in the Institutional Review Board Statement, the Ethics Review Board of University of Valencia approved all the procedures (registration number 1855485), which were all performed in accordance with the principles of the Declaration of Helsinki of the World Medical Association.

Reviewer 2 Report

The authors presented in this study that a single bout of IPC prior to a moderate-intensity walking session has an impact on cardiovascular parameters, by studying on 9 participants. The study in general is solid and meaningful, and I would like to suggest as follows:

1.       I don’t find the information for the equipment and reagents were used for the experiment.

2.       It would be helpful to add the difference between T0 and T1 in Table 1 and 2.

Author Response

Response to reviewers: Potential benefits of a single session of remote ischemic preconditioning and walking in sedentary older adults. A pilot study

Manuscript ID: ijerph-2228162

We thank the reviewers (1 and 2) for their critical review and valuable comments. We have taken into account all of their recommendations and suggestions. Itemized responses are listed below. The modifications have been tracked throughout the manuscript to make its revision easier (See Revised Manuscript).

Reviewer 2:

The authors presented in this study that a single bout of IPC prior to a moderate-intensity walking session has an impact on cardiovascular parameters, by studying on 9 participants. The study in general is solid and meaningful, and I would like to suggest as follows:

We really appreciate the comments and suggestions from the reviewer.

  1. I don’t find the information for the equipment and reagents were used for the experiment.

In the methods section, each of the devices used for each measurement (lines 121, 141, 151) as well as for the interventions (lines 113-114) has been specified.

  1. It would be helpful to add the difference between T0 and T1 in Table 1 and 2.

The reviewer is totally right in his/her suggestion. We have included mean differences between T0 and T1 in tables 1 and 2.
